# Manifold-Preserving Transformers are Effective for Short-Long Range Encoding

**Ayan Sengupta**
IIT Delhi, India
eez228655@ee.iitd.ac.in

**Md. Shad Akhtar**
IIIT Delhi, India
shad.akhtar@iiitd.ac.in

**Tanmoy Chakraborty**
IIT Delhi, India
tanchak@iitd.ac.in

## Abstract

Multi-head self-attention-based Transformers have shown promise in different learning tasks. Albeit these models exhibit significant improvement in understanding short-term and long-term contexts from sequences, encoders of Transformers and their variants fail to preserve layer-wise contextual information. Transformers usually project tokens onto sparse manifolds and fail to preserve mathematical equivalence among the token representations. In this work, we propose `TransJect`, an encoder model that guarantees a theoretical bound for layer-wise distance preservation between a pair of tokens. We propose a simple alternative to dot-product attention to ensure Lipschitz continuity. This allows `TransJect` to learn injective mappings to transform token representations to different manifolds with similar topology and preserve Euclidean distance between every pair of tokens in subsequent layers. Evaluations across multiple benchmark short- and long-sequence classification tasks show maximum improvements of 6.8% and 5.9%, respectively, over the variants of Transformers. Additionally, `TransJect` displays 79% better performance than Transformer on the language modeling task. We further highlight the shortcomings of multi-head self-attention from the statistical physics viewpoint. Although multi-head self-attention was incepted to learn different abstraction levels within the networks, our empirical analyses suggest that different attention heads learn randomly and unorderly. In contrast, `TransJect` adapts a mixture of experts for regularization; these experts are more orderly and balanced and learn different sparse representations from the input sequences. `TransJect` exhibits very low entropy and can be efficiently scaled to larger depths.

## 1 Introduction

Over the past few decades, Deep Neural Networks have greatly improved the performance of various

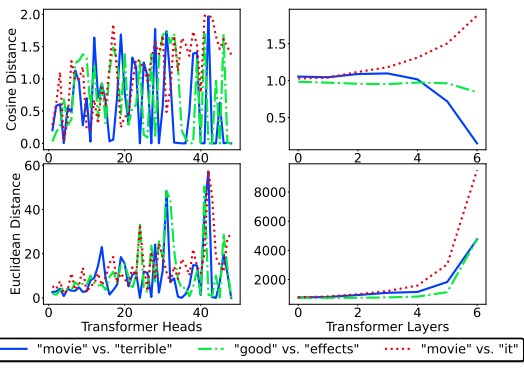

Figure 1: Layer-wise distances between a few selected tokens from the text "*This movie is terrible but it has some good effects.*" We use a trained Transformer model to extract token representations from different layers.

downstream applications. Stacking multiple layers has been proven effective in extracting features at different levels of abstraction, thereby learning more complex patterns (Brightwell et al., 1996; Poole et al., 2016). Since then, tremendous efforts have been made in building larger depth models and in making them faster (Bachlechner et al., 2020; Xiao et al.). Self-attention-based Transformer model (Vaswani et al., 2017) was proposed to parallelize the computation of longer sequences; it has achieved state-of-the-art performance in various sequence modeling tasks. Following this, numerous efforts have been made to reduce computation and make the Transformer model suitable even for longer sequences (Katharopoulos et al., 2020; Peng et al., 2020; Kitaev et al., 2020; Beltagy et al., 2020; Press et al., 2021; Choromanski et al., 2021; Tay et al., 2021). However, very few of these studies discuss information propagation in large-depth models. A recent study (Voita et al., 2019) characterized how Transformer token representations change across layers for different training objectives. The dynamics of layer-wise learning are essential to understand different abstract levels a model could learn, preserve and forget to continue learning throughout the layers. However, recent

studies need to shed more light on how Transformers preserve contextual information across different layers (Wu et al., 2020; Voita et al., 2019). To understand how Transformer encodes contextual similarities between tokens and preserves the similarities layer-wise and across different attention heads, we highlight an example in Figure 1. We select three pairs of semantically-related tokens. We observe the Euclidean and cosine distances of the representations learned at different layers of the Transformer trained on the IMDb sentiment classification task. Although the trained Transformer model preserves the semantic similarity among the same tokens across layers, the Euclidean distance between their representations increases in the upper layers. This indicates that Transformer projects the representations to different and sparse subspaces (a subspace with low density), albeit preserving the angle between them. Additionally, we observe that the distances between different token representations vary across different attention heads at different encoder layers in a haphazard manner. Preserving distance and semantic similarity across layers is vital to ensure the continual learning capabilities of deep neural models, which Transformer evidently fails to do.

Neuroscientists have been working for years to understand how the human brain functions and simulates the behaviours of physical sciences (Koch and Hepp, 2006; Vértes et al., 2012). Arguably, the human brain is more capable of 'associative learning' and 'behavioural formation', which can be attributed to the number of neurons and their inter-connectedness (synapses) rather than the size of the brain, or the number of layers through which the information propagates (Dicke and Roth, 2016). Although a fully-grown human brain can have hundreds of billions of neurons, it has been found that at a time, only a tiny fraction of neurons fire (Ahmed et al., 2020; Poo and Isaacson, 2009). This sparse firing can help the brain sustain low entropy (energy). *Entropy*, a measure that quantifies a state's randomness, has thus become an essential tool to understand the factors behind human intelligence (Saxe et al., 2018; Keshmiri, 2020) and how the human brain operates.

Unfortunately, Transformer and its variants are yet to be studied from these interdisciplinary viewpoints. Our study explores the connection between model sparsity and entropy. Towards this, we propose a complete redesign of self-attention-

based **Trans**former with enforced in**ject**ivity, *aka* `TransJect`. With injectivity, our model imposes the constraint in which the representations of two distinct tokens always differ across all the layers. Unlike Transformer, which only injects regularization through multi-head self-attention and dropout, `TransJect` does not require explicit regularizers and can be regularized implicitly due to its inherent injectivity. The backbone of `TransJect` is a non-normalized linear orthogonal attention and an injective residual connection; both ensure Lipschitz continuity. By enforcing Lipschitz continuity in Euclidean and dot-product space, the model can preserve the manifold structure between tokens and perform well on the final predictive task.

To validate our hypotheses and empirically justify the superiority of our model, we use two short and five long sequence classification tasks. `TransJect` outperforms Transformer and other benchmark variants with an average margin of $3.4\%$ and $2.2\%$ on the short- and long-sequence classification tasks, respectively. Our model performs best on the long sequence classification (LRA) benchmark, achieving $0.2\%$ better accuracy than the best baseline, Skyformer (Chen et al., 2021). We further demonstrate `TransJect` on language modeling task on the Penn TreeBank dataset, in which our model achieves $79\%$ better test perplexity than the vanilla Transformer. Empirical analyses suggest a very low entropy of `TransJect` representations, indicating that `TransJect` captures sparse and more orderly representations compared to Transformer. Moreover, `TransJect` shows $13\times$ lower inference runtime than Transformer, indicating its efficiency in encoding long input sequences.[1]

## 2 Related Works

Despite being a ubiquitous topic of study across different disciplines of deep learning, Transformer (Vaswani et al., 2017) models still require better mathematical formalization. On a recent development, Vuckovic et al. (2020) formalized the inner workings of self-attention maps through the lens of measure theory and established the Lipschitz continuity of self-attention under suitable assumptions. However, the Lipschitz condition depends on the boundedness of the representation space and the Lipschitz bound of the

---

[1]The source codes of `TransJect` can be found at https://github.com/victor7246/TransJect.git.

fully-connected feed-forward layer (FFN). A similar study (Kim et al., 2021) also concluded that the dot-product self-attention is neither Lipschitz nor injective under the standard conditions. Injectivity of the transformation map is essential to ensure that the function is bijective and, therefore, reversible. Reversibility within deep neural networks has always been an active area of study (Gomez et al., 2017; Arora et al., 2015; Chang et al., 2018). A reversible network ensures better scaling in large depths and is more efficient than non-reversible structures. Recently, Mangalam et al. (2022) designed a reversible Transformer and empirically highlighted its effectiveness in several image and video classification tasks. However, any similar development has yet to be made for developing scalable reversible sequential models.

*Dynamical isometry* is a property that mandates the singular values of the input-output Jacobian matrix to be closer to one. Pennington et al. (2017) showed that dynamical isometry could aid faster convergence and efficient learning. Following their idea, Bachlechner et al. (2020) showed that residual connections in Transformers do not often satisfy dynamical isometry, leading to poor signal propagation through the models. To overcome this, they proposed residual with zero initialization (ReZero) and claimed dynamical isometry for developing faster and more efficient large-depth Transformers. Previously, Qi et al. (2020) enforced a stronger condition of *isometry* to develop deep convolution networks efficiently. However, isometric assumptions are not valid for sequential modeling due to different levels of abstraction within the input signals. Moreover, isometry may not hold between contextually-dissimilar tokens.

Another essential aspect behind designing efficient large-depth models is ensuring *model sparsity*. Several notable contributions have been made (Baykal et al., 2022; Jaszczur et al., 2021; Li et al., 2023; Tay et al., 2020a) to enforce sparse activations within Transformers to make them more efficient and scalable. Li et al. (2023) argued that sparse networks often resemble the sparse activations by the human brain, bearing a similarity between artificial and biological networks. They empirically showed that the trained Transformers are inherently sparse, and the sparsity emerges from all the layers. As discussed in the previous section, Transformers project the representations sparsely onto sparse subspaces. Although sparse models are inherently regularized and display lower entropy, projecting the representations onto a sparse subspace pushes them further from being Lipschitz, preventing them from reversible.

This work proposes an injective and Lipschitz continuous alternative to vanilla Transformer, namely, `TransJect`. With the enforced injectivity, our model establishes a theoretical guarantee for reversibility and thus can be scaled to larger depths. Further, `TransJect` displays significantly lower entropy than Transformer, indicating a lower energy footprint and more efficiency.

## 3 Designing Injective Transformer

This section formally describes our proposed model, `TransJect`. It inherits the structure from the vanilla Transformer and achieves a smoother activation plane by utilizing injective maps for transforming token representations across layers. For an $L$-layered stacked encoder, we aim to learn the representation of a sequence $\boldsymbol{X} = \{\boldsymbol{x}_1, \boldsymbol{x}_2, \cdots, \boldsymbol{x}_N\}$ at each layer $l$ that preserves the pairwise distance between every pair of words within a theoretical bound. We illustrate the components of `TransJect` in Figure 2. All the proofs presented in the paper are supplied in Appendix A.

### 3.1 Background

**Activation bound.** For any function $f : \mathbb{R}^n \rightarrow \mathbb{R}^m$, we define the activation bound $K_f$ as $\sup_{\boldsymbol{x} \neq 0} \frac{||\boldsymbol{f}(\boldsymbol{x})||_p}{||\boldsymbol{x}||_p}$, for a suitable integer $p$. A linear map $\boldsymbol{M}$ equals the induced matrix norm $||\boldsymbol{M}||_p$. Intuitively, this is the maximum scale factor by which a mapping expands a vector $\boldsymbol{x}$. In Euclidean space, we usually choose $p = 2$.

**Lipschitz Continuity.** A function $f : \mathbb{R}^n \rightarrow \mathbb{R}^m$ under $||.||_p$ norm is called *Lipschitz continuous* if there exists a real number $K \geq 0$ such that

$$||f(\boldsymbol{x}) - f(\boldsymbol{y})||_p \leq K||\boldsymbol{x} - \boldsymbol{y}||_p. \quad (1)$$

for any $\boldsymbol{x}, \boldsymbol{y} \in \mathbb{R}^n$. Lipschitz continuity can be defined over any metric space, but in this paper, we restrict its definition to only Euclidean space with $p = 2$. $K$ is called *Lipschitz bound*.

### 3.2 Space-Preserving Orthogonal Attention

The backbone of `TransJect` is the space-preserving orthogonal attention.

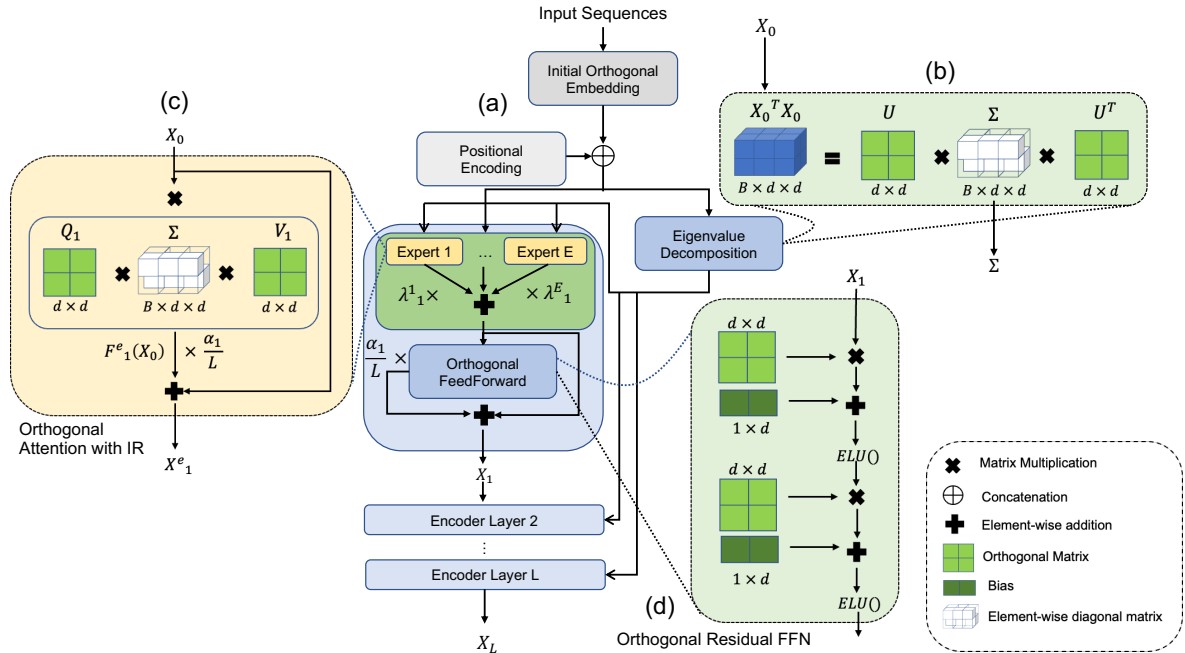

Figure 2: Internals of `TransJect` with (a) an $L$-layered encoder containing a Mixture of Experts (MOE) with $E$ experts, (b) approximated eigenvalue computation, and (c) an orthogonal attention with injective residual. (d) The outputs of the intermediate encoder are fed to the orthogonal residual FFN. The hidden dimension used for representing each token is denoted by $d$.

**Theorem 1** (Space-Preserving Orthogonal Attention). Replacing $\boldsymbol{W}^Q, \boldsymbol{W}^K, \boldsymbol{W}^V$ with real square orthogonal matrices in non-normalized linear self-attention reduces the activation bound to $\sigma_1^2(\boldsymbol{X})$, where $\sigma_1(\boldsymbol{X})$ is the largest singular value of $\boldsymbol{X}$. This reduces the attention operation $Attn(\boldsymbol{X}) = \boldsymbol{X}\boldsymbol{U}\boldsymbol{\Sigma}\boldsymbol{V}$, for learnable orthogonal matrices $\boldsymbol{U}$ and $\boldsymbol{V}$ and the singular values $\boldsymbol{\Sigma}$.

Notice that the activation bound of the modified attention mechanism does not depend on any learnable parameters; instead can be bounded by the largest eigenvalue of $\boldsymbol{X}^T\boldsymbol{X}$. Therefore, we assume a stochastic $\boldsymbol{X}^T\boldsymbol{X}$ to ensure that the largest eigenvalue is always 1 (see Corollary 2 in Appendix A.1), and the attention operator preserves the pairwise distance between any two tokens. We learn orthogonal projection matrices in each layer, $\boldsymbol{U}$ and $\boldsymbol{V}$. In contrast, the diagonal matrix containing eigenvalues $\boldsymbol{\Sigma}$ is learned on the initial embedding obtained from the initial embedding layer defined in Section 3.5, also denoted as $l = 0$. Therefore, in our proposed non-normalized orthogonal linear attention, we compute the attention matrix (encoded in $\boldsymbol{\Sigma}$) only once and learn different projections of it in different layers.

**Approximating eigenvalues.** Eigenvalue decomposition is computationally expensive with a runtime complexity of $\mathcal{O}(Bd^3)$, with $B$ being

the batch size and $d$ being the hidden dimension. This work uses a simple approximation to compute $\tilde{\boldsymbol{\Sigma}}$, the eigenvalues of $\boldsymbol{X}^T\boldsymbol{X}$. Formally, we compute $\tilde{\boldsymbol{U}} = \arg\min_{\boldsymbol{U}} ||\boldsymbol{X}^T\boldsymbol{X} - \boldsymbol{U}\boldsymbol{\Sigma}\boldsymbol{U}^T||$, and $\tilde{\boldsymbol{\Sigma}} = \arg\min_{\boldsymbol{\Sigma}} ||\boldsymbol{X}^T\boldsymbol{X} - \boldsymbol{U}\boldsymbol{\Sigma}\boldsymbol{U}^T||$. To learn the approximate eigenvalues, we can minimize the reconstruction loss $||\boldsymbol{X}^T\boldsymbol{X} - \boldsymbol{U}\boldsymbol{\Sigma}\boldsymbol{U}^T||$ for a learnable orthogonal eigenvector $\boldsymbol{U}$. We use standardization to enforce the stochasticity constraint on $\tilde{\boldsymbol{\Sigma}}$. Further, instead of deriving the eigenvalues, we can also initialize a random diagonal matrix $\tilde{\boldsymbol{\Sigma}}$, without any approximation that optimizes the only task-specific training objective, without enforcing the reconstruction. We denote this version of the model as **Random-TransJect**. We compute $\tilde{\boldsymbol{\Sigma}}$ once, only on the initial token embeddings.

### 3.3 Injective Residual (IR)

We fine-tune the hidden representation for every layer $l$ by learning a new attention projection on the hidden state learned in the previous layer. Formally, we define,

$$\boldsymbol{X}^{(l)} = \boldsymbol{X}^{(l-1)} + \frac{\alpha_l}{L}F(\boldsymbol{X}^{(l-1)}). \quad (2)$$

Here, $F$ is the self-attention operator, followed by a suitable non-linear activation function, and $\alpha_i \in (0, 1)$ is the residual weight. We use a learnable residual weight and a sigmoid activation to

scale it in $(0, 1)$. In the previous studies, ReLU and GELU (Hendrycks and Gimpel, 2016) have been popular choices for the activation function. In this work, we choose ELU (Clevert et al., 2015), a non-linear $\mathcal{C}^1$ (continuous and differentiable) activation function with a Lipschitz bound of 1. Although ReLU is a Lipschitz function with $K = 1$, it is not everywhere differentiable and injective. Following Bachlechner et al. (2020), we adopt ReZero (residual with zero initialization) to enforce dynamical isometry and stable convergence.

**Lemma 1 (Residual contractions are injective).** $f : \boldsymbol{X} \to \boldsymbol{X} + \frac{\alpha_l}{L} F(\boldsymbol{X})$ is injective for $L \geq 3$.

To maintain the dimensionality, Transformer projects the representations to a lower-dimensional space, which reduces the number of synapses among the neurons by a factor of $H$, the number of heads. As opposed to this, we devise a **M**ixture **o**f **E**xpert (MOE) attention (motivated by Shazeer et al. (2017)). With this, we compute $\boldsymbol{X}^{(l,e)}$ for each expert $e \in \{1, 2, \cdots, E\}$ in each layer $l$ using Equation 2, learnable expert weights $\lambda_i^{(l)}$s, and use a convex combination of them to compute,

$$\boldsymbol{X}^{(l)} = \sum_{e=1}^{E} \lambda_e^{(l)} \boldsymbol{X}^{(l,e)}, \quad s.t. \sum_{e=1}^{E} \lambda_e^{(l)} = 1. \quad (3)$$

Note that $\lambda_e^{(l)}$ is computed for each sample, and the same expert weights are used for all tokens within a sample.

**Corollary 1 (Injectivity of MOE).** The mapping function defined in Equation 3 is injective.

### 3.4 Orthogonal Residual FFN (ORF)

We reformulate the position-wise FFNs with orthogonal parameterization. FFN layers in Transformer emulate a key-value memory (Geva et al., 2021). We enforce Lipschitz continuity on the feedforward sublayer to preserve the layer-wise memory. Formally, we define,

$$ORF(\boldsymbol{X}^{(l)}) =$$
$$\boldsymbol{X}^{(l)} + \frac{\alpha_l}{L} ELU\Big(ELU(\boldsymbol{X}^{(l)}\boldsymbol{W}_1 + \boldsymbol{b}_1)\boldsymbol{W}_2 + \boldsymbol{b}_2\Big).$$

With both $\boldsymbol{W}_1$ and $\boldsymbol{W}_2$ being square orthogonal matrices.

**Corollary 2 (Injectivity of ORF).** Orthogonal residual FFNs are injective.
*Proof.* Using the Lipschitz continuity of ELU, we can prove the corollary directly using Lemma 1.

### 3.5 Injective Token Embedding

Transformer introduced conditional encoding to inject tokens' relative and absolute positional information into the self-attention layer. It leverages sinusoidal representations of every position and adds to the original token embeddings to infuse the positional information. To ensure injectivity at each layer, we need to ensure that the initialization of the token embeddings is also injective, *i.e.*, no two tokens should have the same embedding. Unfortunately, the addition operator is not injective. Therefore, we compute the initial embedding of token $x_i$ as $\boldsymbol{X}_i^{(0)} = Concat(Emb(x_i), PE_{i,:})$. Here $PE$ is defined similarly to the positional encoding proposed by Vaswani et al. (2017). The embedding matrix is orthogonally parameterized. Concatenation ensures the injectivity of embeddings. However, to maintain the dimensionality, we learn the initial embedding and positional encoding at a lower dimensional space, $\mathbb{R}^{\frac{d}{2}}$, in which $d$ is the hidden size in the encoder. We define the final encoder mapping for each sublayer $l$ as a composite mapping defined by,

$$SubLayer^{(l)}(\boldsymbol{X}^{(l-1)}) =$$
$$ORF \circ MOE \circ IR(\boldsymbol{X}^{(l-1)}). \quad (4)$$

**Theorem 2.** The composite map defined in Equation 4 is an injective Lipschitz with a fixed (data-independent) upper bound.

A bounded activation bound ensures that the incremental learning of our encoder model reduces with large depth, which makes our model scalable to larger depths. It further enforces the importance of learning better embeddings in the initial embedding layer, which drives the entire encoding. The runtime complexity of our orthogonal non-normalized attention is $\mathcal{O}(Nd^2)$, whereas dot-product self-attention has a runtime complexity of $\mathcal{O}(N^2d + Nd^2)$. In a comparable setting where $N >> d$, `TransJect` should have a lower runtime complexity than Transformer.

## 4 Experimental Setup

### 4.1 Tasks and Datasets

We evaluate `TransJect` and its variants on seven short- and long-sequence classification tasks and one language modeling task. We choose the IMDb movie review sentiment classification (Maas et al., 2011) and the AGnews topic classification (Zhang et al., 2015) datasets for short text classification

| Model | IMDb | AGnews |
|-------|------|--------|
| Transformer | 81.3 | 88.8 |
| Transformer+ReZero | 83.4 | 89.6 |
| Orthogonal Transformer | 85.1 | 86.3 |
| Linformer (Wang et al., 2020)† | 82.8 | 86.5 |
| Synthesizer (Tay et al., 2021)† | 84.6 | 89.1 |
| TransJect | **88.1** | 88.8 |
| Random-TransJect | 86.5 | **90.2** |

Table 1: Text classification accuracy on IMDb and AG-news (results highlighted with † are taken from Tay et al. (2021)).

– the former one is a binary classification task, whereas the latter one contains four classes. To further highlight the effectiveness of TransJect on longer sequences, we evaluate our model on the LRA benchmark (Tay et al., 2020b). LRA benchmark consists of five long sequence classification tasks – ListOps (Nangia and Bowman, 2018), Byte-level text classification on IMDb review (CharIMDb) dataset (Maas et al., 2011), Byte-level document retrieval on AAN dataset (Radev et al., 2013), Pathfinder (Linsley et al., 2018), and Image classification on the CIFAR-10 dataset (Krizhevsky and Hinton, 2010). For language modeling, we use Penn TreeBank (PTB) (Mikolov et al., 2010) dataset, containing 100M tokens. We provide the details of hyperparameter in Appendix B.1.

## 4.2 Results

**Short sequence classification.** Table 1 shows the performance of the competing models. On IMDb classification, TransJect outperforms Transformer with a $6.8\%$ margin. TransJect achieves $3.5\%$ better accuracy than Synthesizer, the best baseline. With zero residual initialization, Transformer can achieve $2.1\%$ better accuracy in the IMDb classification task. We use an additional ablation of Transformer, in which all the learnable weight matrices are parameterized orthogonally. This improves accuracy on IMDb classification by $3.8\%$. On the AGnews topic classification task, Random-TransJect achieves $90.2\%$ accuracy, $1.1\%$ better than the best baseline. Interestingly, Random-TransJect performs better than TransJect on the AGnews classification task; injecting randomness through randomly initialized eigenvalues aids in $1.4\%$ performance improvement. Limited contextual information can create difficulty reconstructing $X^T X$ from the approximate eigenvalues. Therefore, having randomly-initialized eigenvalues can aid in learning better

context when the context itself is limited. To highlight the effectiveness of the MOE module, we evaluate an ablation of our model by dropping the MOE module (reducing the number of experts to 1). Dropping the MOE module reduces the validation accuracy on the IMDb classification task by $4.7\%$. On the other hand, using only a single head in the original Transformer model reduces its performance on the IMDb classification task by $2.1\%$.

**Long sequence classification.** We evaluate TransJect against Transformer along with several of its recent variants and report the test accuracy in Table 2. Similar to short sequence classification tasks, TransJect is very effective for long sequences and consistently outperforms all the baselines. Out of five tasks, TransJect achieves the best performance in three and beats the best baseline, Skyformer by $0.2\%$ on the leaderboard. TransJect achieves $2.3\%$ and $0.2\%$ better test accuracies on ListOps and byte-level text classification tasks than the corresponding best baselines, Big Bird and Skyformer, respectively. Interestingly, Random-TransJect achieves the best performance on the Pathfinder task, with a wide margin of $1.9\%$. The ListOps task evaluates the ability to learn long-range hierarchical dependencies, whereas the Pathfinder task evaluates the ability to learn spatial dependencies. As argued by Chen et al. (2021), learning both these dimensions poses difficulty for self-attention-based methods. With a superior performance on both tasks, TransJect showcases its effectiveness in learning long-term patterns from both temporal sequences and sequences with different hierarchical dimensions. Moreover, Random-TransJect performs better than TransJect on both Pathfinder and Image classification tasks with a margin of $1.1\%$ and $1.3\%$, respectively. We argue that the sparsity in inputs in these two visual tasks is difficult to be approximated by the eigenvalues, which costs TransJect in these tasks.

**Language modeling.** We report validation and test perplexity on PTB in Table 3 for TransJect and other baselines. Our model achieves $79\%$ lower test perplexity than the vanilla Transformer. As argued by Bachlechner et al. (2020), loss-free information propagation is required for training large depth models. Due to this, Transformer with ReZero initialization achieves $75\%$ better performance than the vanilla Transformer that uses uni-

| Model | ListOps | Text | Retrieval | Pathfinder | Image | Avg. |
|---|---|---|---|---|---|---|
| Transformer | 38.4 | 61.9 | 80.7 | 65.2 | 40.6 | 57.4 |
| Transformer+ReZero | 38.3 | 59.1 | 79.2 | 68.3 | 38.6 | 56.7 |
| Orthogonal Transformer | 39.6 | 58.1 | 81.4 | 68.9 | 42.2 | 58.0 |
| Reformer (Kitaev et al., 2020)† | 37.7 | 62.9 | 79.0 | 66.5 | **48.9** | 59.0 |
| Big Bird (Zaheer et al., 2020)† | 39.3 | 63.9 | 80.3 | 68.7 | 43.2 | 59.1 |
| Linformer (Wang et al., 2020)† | 37.4 | 58.9 | 78.2 | 60.9 | 38.0 | 54.7 |
| Informer (Zhou et al., 2021)† | 32.5 | 62.6 | 77.6 | 57.8 | 38.1 | 53.7 |
| Nystromformer (Xiong et al., 2021)† | 38.5 | 64.8 | 80.5 | 69.5 | 41.3 | 58.9 |
| Performers (Choromanski et al., 2021)† | 38.0 | 64.2 | 80.0 | 66.3 | 41.4 | 58.0 |
| Skyformer (Chen et al., 2021)† | 38.7 | 64.7 | **82.1** | 70.7 | 40.8 | 59.4 |
| TransJect | **42.2** | **64.9** | 80.3 | 71.5 | 38.9 | **59.6** |
| Random-TransJect | 40.1 | 64.6 | 80.2 | **72.6** | 40.2 | 59.5 |

Table 2: Test accuracy on the LRA benchmark (results highlighted with † are taken from Chen et al. (2021)).

| Model | Val | Test |
|---|---|---|
| Transformer | 636.55 | 598.93 |
| Transformer+ReZero | 161.52 | 147.80 |
| Orthogonal Transformer | 179.98 | 165.78 |
| TransJect | **134.82** | **127.59** |
| Random-TransJect | 215.68 | 199.72 |

Table 3: Perplexity (lower value is better) calculated for language modeling on PTB dataset on validation ('Val') and testing ('Test') splits.

form residual weights. However, it is worth noting that TransJect achieves $14\%$ better performance than ReZero initialization, which could be attributed to the inherent injectivity, that ReZero fails to ensure. On the other hand, TransJect with random eigenvalues performs poorly due to its inability to encode the inter-dependence between tokens.

## 5 Analysis

To understand the connections behind model depth, activation bounds and entropies, we conduct detailed statistical analyses on our model and Transformer. We use the IMDb and CharIMDb classification tasks for these studies as part of short long-range classification tasks, respectively. We use the outputs inferred by our models on a subsample of the test data for these analyses.

**Activation bounds and entropy.** Continuing our initial discussion on preserving layer-wise distances between tokens, we calculate the distribution of activation bounds at different encoding layers. As defined in Section 3.1, we compute the *activation factor* for each encoding layer for TransJect and Transformer. Formally, for $l^{th}$ layer, we compute the activation factor

$$\mathbb{A}^{(l)} = \mathbb{E}_X \mathbb{E}_{i \neq j} \left[ \frac{||X_i^{(l)} - X_j^{(l)}||}{||X_i^{(0)} - X_j^{(0)}||} \right]. \quad (5)$$

Here $X^{(l)} \in \mathbb{R}^{N \times d}$ is the hidden representation of the sequence $X$ at $l^{th}$ layer.

Similarly, we compare the *differential entropy* (*aka* entropy) of the hidden representations learned by TransJect at each layer to understand how the entropy state changes across the layers. We calculate differential entropy as,

$$entropy^{(l)}\left(X^{(l)}\right) = \mathbb{E}_{j,h}\left[-\log P(X_{j,h}^{(l)})\right]$$
$$= -\mathbb{E}_j\left[\int_{\mathbb{H}} P(X_{j,h}^{(l)}) \log P(X_{j,h}^{(l)}) d\boldsymbol{h}\right] \quad (6)$$

where, $P$ is the empirical probability density function of $X$. Note that $X_{j,h_i}^{(l)} \underset{h_i \neq h_j}{=} X_{j,h_j}^{(l)}$ leads the entropy to $-\infty$. Therefore, sparser states always have lower entropy and are more deterministic. At the same time, unorderly, random and stochastic states have higher entropy. We highlight the distribution of activation factors and entropy of token embeddings at every layer of TransJect and Transformer in Figure 3a. Under Euclidean and dot-product, TransJect displays an empirical activation bound of $\approx 1$. Unlike TransJect, Transformer has much higher empirical activation bounds. Although orthogonal parameterization and ReZero lead to much lower activation bounds, they are still higher than our model. Interestingly, Transformer aims to preserve the semantic similarity at the later layers at the expense of distance; however, TransJect can preserve both of them with a tighter bound, leading to a more robust representation for each token. We hypothesize that restricting the distance between a pair of tokens acts as a regularization, improving the encoder's final predictive

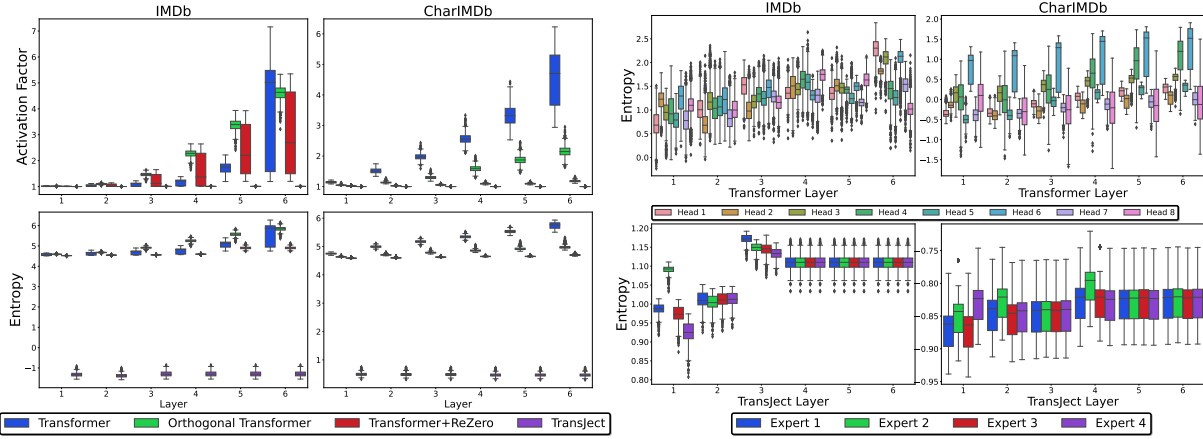

(a) Distribution of activation and entropy of models on IMDb and CharIMDb classification tasks.

(b) Attention head/expert level distribution of entropy.

Figure 3: We observe higher activation factors for Transformer, whereas the median empirical activation factor for `TransJect` is $\approx 1$. Lower entropy for `TransJect` indicates that at the neuron level as well as at the expert level, the representations are more orderly.

performance. We computed the Pearson's correlation between the activation bound of different encoder models (TransJect, Transformer, Orthogonal Transformer and Transformer+ReZero) and the final predictive performance and obtained a correlation score of $-0.81$ with pvalue $0.09$. A similar correlation score of $-0.68$ is obtained on the CharIMDb classification task. These results highlight the negative linear relationship between the final predictive performance and the average activation bound of the encoders. Therefore, we conclude that a tighter activation bound could lead to better predictive performances on short- and long-range encoding tasks.

The average entropy obtained by `TransJect` on IMDb classification is $-1.3$ with a standard deviation of $0.12$. On the other hand, Transformer obtains an average entropy of $5.80$ with a standard deviation of $0.13$. With a higher activation factor, Transformer projects the tokens onto much sparser and random subspaces, increasing the system's overall entropy. On the other hand, representations learned by `TransJect` are more orderly and thus have low entropy. Moreover, the entropy of `TransJect` does not increase perpetually, which according to the second law of thermodynamics, suggests a reversible process. We compute Spearman's rank and Pearson correlation to understand the relationship between average activation bound and average entropy. We observe a Spearman's rank correlation value of $1.0$ on the IMDb classification task, with a p-value of $0.001$. The Pearson's correlation also stands at $0.69$. These analyses indicate the pos-

itive relationships between the two measures, i.e. having a lower activation bound lowers the overall entropy of the model representations. The same argument can be used to explain the higher entropy in later layers of the Transformer. Our analyses also confirm the positive correlation between model depth, activation and entropy for Transformer(see Figure 6 in Appendix C). It is noteworthy that dot-product self-attention computes the query-key similarity between different tokens. Contrarily, in our proposed attention mapping, we compute the similarities between neurons *i.e.*, similarities between different projection vectors, which in turn, enforces our model to learn different projection dimensions and reduces the overall entropy of the system. Additionally, we report the entropy of the representations at each attention head and expert level in Figure 3b. Although `TransJect` displays higher entropy at the expert level, it is still lesser than the corresponding attention heads of the Transformer model. A sparse load-balancing capability of the expert model (see Figure 7 in Appendix C) ensures lower entropy throughout the model training. Further, the changes in inter-quartile ranges in the later layers of the Transformer show increasing randomness in larger depths, which can also be attributed to the higher activation factors. On the other hand, `TransJect` stabilises the entropy in the later layers.

**Preserving distances between tokens.** Figure 4 shows representations obtained on tokens of a sample text at different encoder layers, projected onto 2-D. We use isometric mapping (Tenenbaum et al., 2000) for projecting the high dimensional

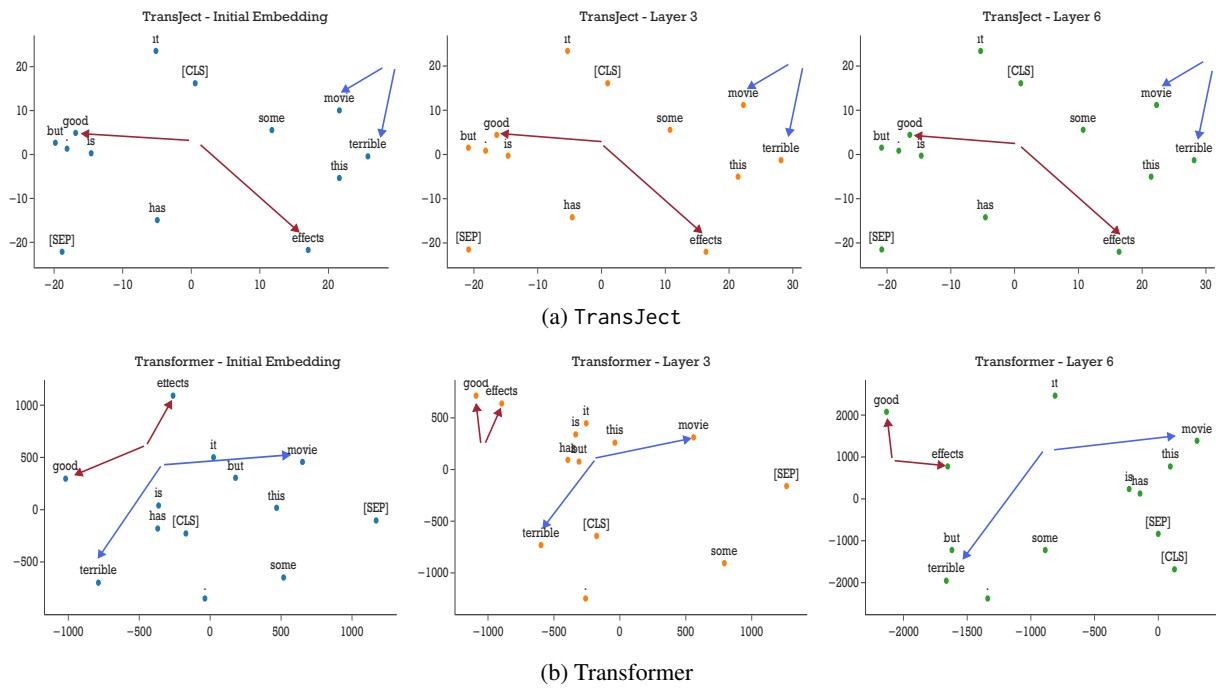

(a) TransJect

(b) Transformer

Figure 4: Isomap plot of layer-wise token embeddings learned by (a) TransJect and (b) Transformer on the text "*This movie is terrible, but it has some good effects.*" We highlight four semantically-relevant words and visualize their positions to understand how different models project tokens onto different subspaces in different encoding layers. TransJect preserves the relative distances between the tokens and their interconnectedness over the layers, indicating that the projection manifolds are topologically similar.

| Model | Speed ↑ | | | |
|---|---|---|---|---|
| | $1k$ | $2k$ | $3k$ | $4k$ |
| Transformer | 1.0 | 1.0 | 1.0 | 1.0 |
| Transformer+ReZero | 1.0 | 1.0 | 1.0 | 1.1 |
| Orthogonal Transformer | 1.0 | 1.0 | 1.1 | 1.4 |
| TransJect | **5.0** | **9.6** | **12.8** | **26.3** |

Table 4: Test-time inference speed on long-range text classification tasks on various text lengths. The reported numbers are speedups w.r.t. Transformers.

vectors to the $2-$D space. TransJect maintains the initial embedding space throughout the layers, showing robustness in learning initial embeddings. On the other hand, Transformer expands the projection subspaces to more sparse subspaces, even though they project semantically similar tokens closer.

**Efficiency comparison.** We report the test-time speed on the CharIMDb classification task with different lengths of input sequences in Table 4. Albeit having $50\%$ more parameters (see Table 5 in Appendix C) than vanilla Transformer, on average, we observe $13\times$ speedup for TransJect, which increases to $26\times$ for longer sequences. From the definitions of thermodynamics, a higher entropy leads an irreversible process. This means that a

model with a high activation bound is more irreversible and, therefore, be less efficient. On the other hand, TransJect exhibits lower entropy, and has higher available energy (from principles of thermodynamics), leading to more efficiency.

## 6 Conclusion

In this work, we introduced TransJect, a new learning paradigm in language understanding by enforcing a distance-preserving criterion in the multi-layer encoder models. We derived that by enforcing orthogonal parameterization and utilizing smoother activation maps, TransJect can preserve layer-wise information propagation within a theoretical bound, allowing the models to regularize inherently. We further argued in favor of injectivity and Lipschitz continuity for better generalization and efficiency. Our empirical analyses suggested a superior performance of TransJect over other self-attention-based baselines. We observed lower entropy with TransJect with low variance, confirming the reversible process of statistical mechanics. These findings will encourage practitioners to explore the natural laws of science for building better, more intuitive, and cognitive AI.

# 7 Ethical Considerations

**Limitations.** As discussed previously, `TransJect` uses approximated eigenvalues to approximate the gram matrix $X^T X$. Therefore, our proposed model could be ineffective for sequences with limited context. Additionally, our model usually exhibits higher forward pass time than non-parameterized models due to orthogonal parametrization.

**Intended Use and Broader Impact.** The empirical analyses and insights gathered in this work encourage us to explore the dependence between sparsity, entropy and model representations. Our study can be further utilized in developing more efficient larger-depth language models.

**User Privacy.** There is no known user privacy concern with this work.

**Potential Misuse.** There is no known potential misuse of this work.

**Reproducibility.** We furnish the experimental details Appendix B.1 for reproducing the results reported in this paper. We have open-sourced the source code at `https://github.com/victor7246/TransJect.git`.

**Environmental Impact.** On the language modeling task, each training iteration takes $\sim 0.6$ seconds on a single Tesla T4 GPU. Total CO2 emissions are estimated to be 0.06 kg. Estimations were conducted using the MachineLearning Impact calculator presented in Lacoste et al. (2019).

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

## A   Theoretical Results

### A.1   Background

We furnish the background materials and proofs of all the theoretical results presented in the main text.

**Lemma 2** (Activation bound of linear maps) For a matrix $M \in \mathbb{R}^{n \times m}$, $K_M$ is same as the largest absolute singular value, under $||.||_2$.

*Proof.*

$$K_M = \sup_{x \neq 0} \frac{||Mx||_2}{||x||_2} = \sup_{||x||_2 = 1} ||Mx||_2.$$

Squaring both the side, we decompose $M$ as $U \Sigma V^T$, where $U$ and $V$ are orthogonal matrices, and $\Sigma$ is the diagonal matrix containing the singular values (square root of eigenvalues of $M^T M$), $\Sigma_1 \geq \Sigma_2 \geq \Sigma_3 \cdots \geq 0$. For an orthogonal matrix $U$, $||Ux||_2^2 = ||x^T U^T U x||_2 = ||x^T x||_2 = ||x||_2^2$. This leads to

$$||Mx||_2^2 = ||U \Sigma V^T x||_2^2 = ||\Sigma x||_2^2$$
$$= \sum_{i=1}^{n} \Sigma_i^2 x_i^2.$$

Hence,

$$K_M^2 = \sup_{\sum_{i=1}^{n} x_i^2 = 1} \sum_{i=1}^{n} \Sigma_i^2 x_i^2.$$

Being a convex sum, $K_M^2 \leq \Sigma_1^2$, which completes the proof.

**Corollary 2** (Largest eigenvalue of a square stochastic matrix) The largest absolute value of any eigenvalue of a square stochastic matrix is equal to $1$.

*Proof.* For any square stochastic matrix $M \in \mathbb{R}^{n \times n}$,

$$MI = \begin{pmatrix} \sum_{j=1}^{n} M_{j,1} \\ \sum_{j=1}^{n} M_{j,2} \\ \vdots \\ \sum_{j=1}^{n} M_{j,n} \end{pmatrix} = \begin{pmatrix} 1 \\ 1 \\ \vdots \\ 1 \end{pmatrix} = I.$$

Hence, 1 is an eigenvalue of $M$. Next, we prove that 1 is the largest eigenvalue of $M$. For any eigenvalue $\lambda$ and its corresponding eigenvector $\boldsymbol{v}$, $\lambda \boldsymbol{v} = \boldsymbol{M}\boldsymbol{v}$. Without loss of generality, we assume $\arg\max_i |v_i| = 1$. Hence, for the 1st entry in this column vector $\lambda v_1 = \sum_{j=1}^n \boldsymbol{M}_{1,j} v_j$. Using triangle inequality we get,

$$|\lambda||v_1| \le |\lambda v_1| = |\sum_{j=1}^n \boldsymbol{M}_{1,j} v_j| \le \sum_{j=1}^n |\boldsymbol{M}_{1,j}||v_1|$$
$$\le |v_1|.$$

Hence, for any eigenvalue $|\lambda| \le 1$. Hence, it proves our corollary that the largest eigenvalue is 1.

**Lemma 3** (Lipschitz bound for continuously differentiable functions). Any $\mathcal{C}^1$ function $f : \mathbb{R}^n \to \mathbb{R}^m$ with bounded derivative has Lipschitz bound as $sup_{\boldsymbol{x}}||\nabla_{\boldsymbol{x}} f||$.

*Proof.* For any $\boldsymbol{x}, \boldsymbol{y} \in \mathbb{R}^n$, we define $g : [0,1] \to \mathbb{R}^m$ as

$$g(\boldsymbol{t}) = f(\boldsymbol{x} + \boldsymbol{t}(\boldsymbol{y} - \boldsymbol{x})). \qquad (7)$$

It is easy to verify that $g(0) = f(\boldsymbol{x})$ and $g(1) = f(\boldsymbol{y})$.

$$f(\boldsymbol{y}) - f(\boldsymbol{x}) = g(1) - g(0) = \int_0^1 \nabla_{\boldsymbol{t}} g(\boldsymbol{t}) d\boldsymbol{t}. \qquad (8)$$

Using chain rule of differentiation on Equation 7, we get,

$$\nabla_{\boldsymbol{t}} g(\boldsymbol{t}) = \nabla_{\boldsymbol{t}} f\Big(x + \boldsymbol{t}(y - x)\Big)(y - x).$$

Using this in Equation 8, we get

$$||f(\boldsymbol{y}) - f(\boldsymbol{x})||$$
$$= \Big|\Big|\int_0^1 \nabla_{\boldsymbol{t}} f\Big(\boldsymbol{x} + \boldsymbol{t}(\boldsymbol{y} - \boldsymbol{x})\Big)(\boldsymbol{y} - \boldsymbol{x}) d\boldsymbol{t}\Big|\Big|$$
$$\le \int_0^1 \Big|\Big|\nabla_{\boldsymbol{t}} f\Big(\boldsymbol{x} + \boldsymbol{t}(\boldsymbol{y} - \boldsymbol{x})\Big)(\boldsymbol{y} - \boldsymbol{x}) d\boldsymbol{t}\Big|\Big|.$$

As $f$ is $\mathcal{C}^1$, the supremum of its derivative exists, allowing us to set $sup_{\boldsymbol{t}}\Big|\Big|\nabla_{\boldsymbol{t}} f\Big(\boldsymbol{x} + \boldsymbol{t}(\boldsymbol{y} - \boldsymbol{x})\Big)\Big|\Big| = K$ and deduce

$$||f(\boldsymbol{y}) - f(\boldsymbol{x})|| \le K||\boldsymbol{y} - \boldsymbol{x}|| \int_0^1 d\boldsymbol{t} = K||\boldsymbol{y} - \boldsymbol{x}||.$$

## A.2 Main Results

## A.3 Proof of Theorem 1

Here we use the linear attention (Katharopoulos et al., 2020) with the dot-product similarity between $\boldsymbol{Q}$ and $\boldsymbol{K}$. Being a real symmetric matrix, $\boldsymbol{X}^T\boldsymbol{X}$ can be decomposed into $\tilde{Q}\Sigma\tilde{Q}^T$, which leads to

$$Attention(\boldsymbol{Q}, \boldsymbol{K}, \boldsymbol{V}) =$$
$$\boldsymbol{X} \underbrace{\boldsymbol{W}^Q \boldsymbol{W}^{K^T} \tilde{Q}}_{\text{orthogonal}} \underbrace{\Sigma}_{\text{diagonal}} \underbrace{\tilde{Q}^T \boldsymbol{W}^V}_{\text{orthogonal}}. \qquad (9)$$

As the product of two orthogonal matrices is orthogonal, Equation 9 reduces to

$$Attention(\boldsymbol{Q}, \boldsymbol{K}, \boldsymbol{V}) = \boldsymbol{X}\boldsymbol{U}\Sigma\boldsymbol{V} \qquad (10)$$

with a suitable set of learnable orthogonal matrices $\boldsymbol{U}$ and $\boldsymbol{V}$, and $\Sigma$ being the matrix containing the eigenvalues of $\boldsymbol{X}^T\boldsymbol{X}$.

## A.4 Proof of Lemma 1

Let us assume $\exists \boldsymbol{x} \ne \boldsymbol{y}$ such that $f(\boldsymbol{x}) = f(\boldsymbol{y})$, which implies $||f(\boldsymbol{x}) - f(\boldsymbol{y})|| = 0$. Using triangle inequality and Equation 2 we get

$$\boldsymbol{x} - \boldsymbol{y} = f(\boldsymbol{x}) - f(\boldsymbol{y}) - \frac{\alpha_l}{L} \cdot (F(\boldsymbol{x}) - F(\boldsymbol{y}))$$

This implies,

$$||\boldsymbol{x} - \boldsymbol{y}||$$
$$\le ||f(\boldsymbol{x}) - f(\boldsymbol{y})|| + |-\frac{\alpha_l}{L}| \cdot ||F(\boldsymbol{x}) - F(\boldsymbol{y})||.$$

Therefore,

$$||\boldsymbol{x} - \boldsymbol{y}|| \le |\frac{\alpha_l}{L}| \cdot ||F(\boldsymbol{x}) - F(\boldsymbol{y})||$$
$$< \frac{1}{L} \cdot ||F(\boldsymbol{x}) - F(\boldsymbol{y})||.$$

Here $F(\boldsymbol{X}) = ELU(\boldsymbol{X}\boldsymbol{W}\boldsymbol{X}^T\boldsymbol{X}\boldsymbol{W}^V)$, for a suitable choice of orthogonal weights $\boldsymbol{W}$ and $\boldsymbol{W}^V$. Although it is difficult to analytically calculate the matrix derivative of $\boldsymbol{X}\boldsymbol{W}\boldsymbol{X}^T\boldsymbol{X}\boldsymbol{W}^V$ w.r.t. $\boldsymbol{X}$, intuitively, we can calculate that $\boldsymbol{X}\boldsymbol{W}\boldsymbol{X}^T\boldsymbol{X}\boldsymbol{W}^V \approx \mathcal{O}(\boldsymbol{X}\boldsymbol{X}^T\boldsymbol{X})$, and thus the norm of the input-output Jacobian is bounded by $3 \cdot ||\boldsymbol{X}||^2$. Here we can ignore the weight matrices, as they are orthogonally parameterized and therefore has $||.|| = 1$. As $\boldsymbol{X}^T\boldsymbol{X}$ is stochastic, the norm of the input-output

Jacobian is bounded by 3. Using Lemma 3, we can calculate the Lipschitz bound of ELU activation as 1. Therefore, using the chain rule of calculus and Lemma 3, we can deduce the Lipschitz bound for $F$ as 3. Using this, we get

$$||\boldsymbol{x} - \boldsymbol{y}|| < \frac{3}{L} \cdot ||\boldsymbol{x} - \boldsymbol{y}||.$$

This contradicts that $\frac{3}{L} \leq 1$ for an encoder with $L \geq 3$. Hence, we prove the lemma by contradiction.

### A.5   Proof of Corollary 1

Let us assume $\boldsymbol{x}^{(l)} \neq \boldsymbol{y}^{(l)}$ such that

$$\sum_{e=1}^{E} \lambda_e \boldsymbol{x}^{(l-1)} + \frac{\alpha_l}{L} \sum_{e=1}^{E} \lambda_e F_e(\boldsymbol{x}^{(l-1)})$$
$$= \sum_{e=1}^{E} \lambda_e \boldsymbol{y}^{(l-1)} + \frac{\alpha_l}{L} \sum_{e=1}^{E} \lambda_e F_e(\boldsymbol{y}^{(l-1)}).$$

Using $\sum_{e=1}^{E} \lambda_e = 1$ we obtain,

$$||\boldsymbol{x}^{(l-1)} - \boldsymbol{y}^{(l-1)}||$$
$$= ||\frac{\alpha_l}{L} \sum_{e=1}^{E} \lambda_e F_e(\boldsymbol{x}^{(l-1)}) - \frac{\alpha_l}{L} \sum_{e=1}^{E} \lambda_e F_e(\boldsymbol{y}^{(l-1)})||.$$

Therefore,

$$||\boldsymbol{x}^{(l-1)} - \boldsymbol{y}^{(l-1)}||$$
$$< \frac{1}{L} || \sum_{e=1}^{E} \lambda_e F_e(\boldsymbol{x}^{(l-1)}) - \sum_{e=1}^{E} \lambda_e F_e(\boldsymbol{y}^{(l-1)})||.$$

Using Lipschitz bound of $F_e(\boldsymbol{x}^{(l-1)})$ as 3, we obtain

$$||\boldsymbol{x}^{(l-1)} - \boldsymbol{y}^{(l-1)}|| < \frac{3}{L} \sum_{e=1}^{E} \lambda_e ||\boldsymbol{x}^{(l-1)} - \boldsymbol{y}^{(l-1)}||$$
$$= \frac{3}{L} ||\boldsymbol{x}^{(l-1)} - \boldsymbol{y}^{(l-1)}||.$$

This contradicts that fact that $\frac{3}{L} \leq 1$. Hence, we prove the corollary by contradiction.

### A.6   Proof of Theorem 2

The original input space of the composite mapping is the concatenated token embedding that lies in $[-1, 1]^d$, as both orthogonal embedding and position encoding functions have the function image (codomain) in $[-1, 1]^{\frac{d}{2}}$. As the input space is compact, any continuously differentiable function is Lipschitz. Therefore, the composite function is automatically Lipschitz continuous. Using Lemma 3 and the fact that the composition of multiple injective functions is injective, we can deduce that the encoding function is injective for any encoder with $L \geq 3$. The dynamical isometry property can also prove the Lipschitz continuity of the encoding function irrespective of the number of encoding layers. Although ReZero only guarantees dynamical isometry during initialization, we can observe the distribution of residual weights throughout model training. We highlight the distribution of residual weights of `TransJect` for IMDb and CharIMDb classification tasks in Figure 5. Residual weight values remain $< 0.33$, highlighting that the residual connections remain injective.

Next, we compute the Lipschitz bound for $f$. For the sake of simplicity, let us assume $\frac{\alpha_1}{L} = \frac{\alpha_2}{L} \cdots = \frac{\alpha_l}{L} = \alpha < \frac{1}{L}$. We first expand $f$ as

$$f(\boldsymbol{x}) = \sum_{e=1}^{E} \lambda_e \boldsymbol{x} + \sum_{e=1}^{E} \lambda_e \alpha \cdot ELU(\boldsymbol{x} \boldsymbol{U}^e \Sigma \boldsymbol{V}^e)$$
$$+ \alpha \cdot ELU(\overline{\boldsymbol{x}}).$$

$$\overline{\boldsymbol{x}} = ELU \Big( \sum_{e=1}^{E} \lambda_e \boldsymbol{x} \boldsymbol{W}_1 +$$
$$\sum_{e=1}^{E} \lambda_e \alpha \cdot ELU(\boldsymbol{x} \boldsymbol{U}^e \Sigma \boldsymbol{V}^e) \boldsymbol{W}_1 + \boldsymbol{b}_1 \Big) \boldsymbol{W}_2 + \boldsymbol{b}_2.$$

This implies,

$$\overline{\boldsymbol{x}} = ELU \Big( \boldsymbol{x} \boldsymbol{W}_1 +$$
$$\sum_{e=1}^{E} \lambda_e \alpha \cdot ELU(\boldsymbol{x} \boldsymbol{U}^e \Sigma \boldsymbol{V}^e) \boldsymbol{W}_1 + \boldsymbol{b}_1 \Big) \boldsymbol{W}_2 + \boldsymbol{b}_2.$$

Using the Lipschitz bound of $F(\boldsymbol{X})$ from the proof of Lemma 1 and the fact $\sum_{e=1}^{E} \lambda_e = 1$, we

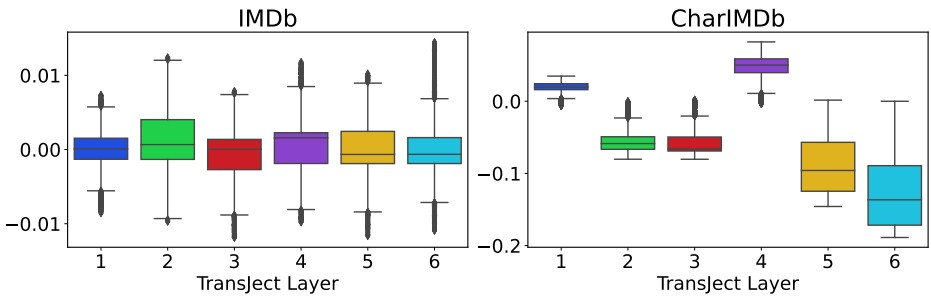

Figure 5: Distribution of Residual Weights in different layers of `TransJect`

get

$$||\overline{\boldsymbol{x}} - \overline{\boldsymbol{y}}||$$
$$\leq (||\boldsymbol{x} - \boldsymbol{y}|| \cdot ||\boldsymbol{W}_1|| + 3|\alpha| \cdot ||\boldsymbol{x} - \boldsymbol{y}||) \cdot ||\boldsymbol{W}_2||$$
$$\leq (1 + 3|\alpha|)||\boldsymbol{x} - \boldsymbol{y}||.$$

Finally,

$$||f(\boldsymbol{x}) - f(\boldsymbol{y})||$$
$$\leq ||\boldsymbol{x} - \boldsymbol{y}|| + 3|\alpha| \cdot ||\boldsymbol{x} - \boldsymbol{y}||$$
$$+ 3|\alpha|(1 + 3|\alpha|) \cdot ||\boldsymbol{x} - \boldsymbol{y}||$$
$$\leq (1 + 3|\alpha|)^2||\boldsymbol{x} - \boldsymbol{y}||$$
$$< (1 + \frac{3}{L})^2||\boldsymbol{x} - \boldsymbol{y}||.$$

Here, $f$ is a layer-wise operator. Hence, for all the $L$ encoder layers,

$$||f(\boldsymbol{x}) - f(\boldsymbol{y})|| < (1 + \frac{3}{L})^{2L}||\boldsymbol{x} - \boldsymbol{y}||.$$

Using $\lim_{n \to \infty}(1 + \frac{1}{n})^n = e$, we get $||f(\boldsymbol{x}) - f(\boldsymbol{y})|| < e^6||\boldsymbol{x} - \boldsymbol{y}||$. Although the theoretical Lipschitz bound is very high, the empirical Lipschitz bound is much smaller due to dynamical isometry. The Lipschitz bound is also independent of the input data, making it theoretically valid. However, as demonstrated in Figure 5, the residual weights are empirically closer to 0, even during the model training, due to which the empirical Lipschitz bound is $\approx 1 << e^6$.

## B  Experimental Setup

### B.1  Hyperparameter settings

For IMDb and AGnews classifications, we choose a maximum text length of $512$. In all these two classification tasks, we keep the configuration of our models with $L = 6$, $E = 4$, and $d = 512$. For these two tasks, we use a mean pooling on the hidden representation obtained by the final encoder layer before passing it to the final classification layer. We utilize the BERT pre-trained tokenizer[2] to tokenize the texts for these two tasks. We follow the evaluation methodologies followed by Chen et al. (2021). In all these classification tasks, we keep the configuration of our models with $L = 2$, $E = 4$, and $d = 512$. Except for the text classification task, we use max pooling on the hidden representation obtained from the final encoder layer before feeding to the final classification feed-forward layer. For the PTB language modeling task, we use a maximum sequence length of 35 in both the input and the output. For all the models, we set $L = 6$, and $d = 512$. All IMDb and AGNews classification experiments are run for 30 epochs. We use an early stopping based on the test loss with the patience of 4 epochs to terminate learning on plateaus. For both these experiments, we use Adam optimizer with a learning rate of 0.0005, $\beta_1 = 0.9, \beta_2 = 0.98$ and $\epsilon = 10^{-9}$. We use both training and test batches of size 32. We follow the implementation details shared by Chen et al. (2021) for the LRA benchmark. All these experiments are run for $50k$ steps. We use Adam optimizer with a learning rate of 0.01 for the language modeling task and trained the models for 10 epochs. One Tesla T4 and one Tesla V100 GPU are used for conducting all the experiments. For each task, we report the average performance across three different runs.

To enforce orthogonality during the forward pass and after backpropagation, we use PyTorch parameterization[3]. This implementation uses different orthogonal maps (e.g. householder or Cayley mapping) to ensure orthogonality. It further uses the framework of dynamic trivialization (Lezcano Casado, 2019) to ensure orthogonality after gradient descent. We use the implementation by

---

[2] https://huggingface.co/bert-base-uncased
[3] https://pytorch.org/docs/stable/generated/torch.nn.utils.parametrizations.orthogonal.html

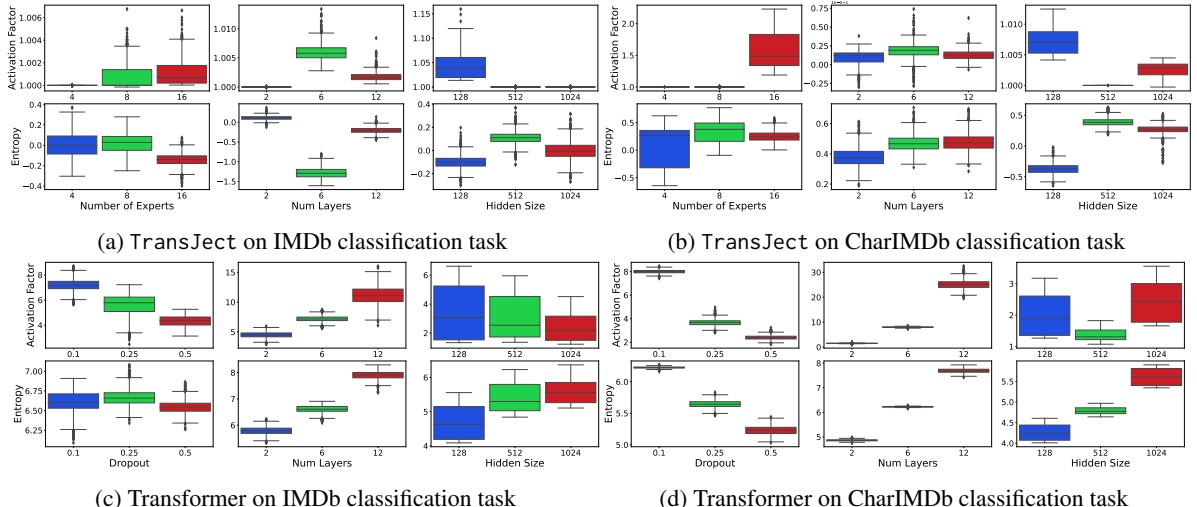

(a) TransJect on IMDb classification task     (b) TransJect on CharIMDb classification task

(c) Transformer on IMDb classification task     (d) Transformer on CharIMDb classification task

Figure 6: Distribution of activation factors and entropy with different model configurations of TransJect and Transformer on IMDb and CharIMDb classification tasks.

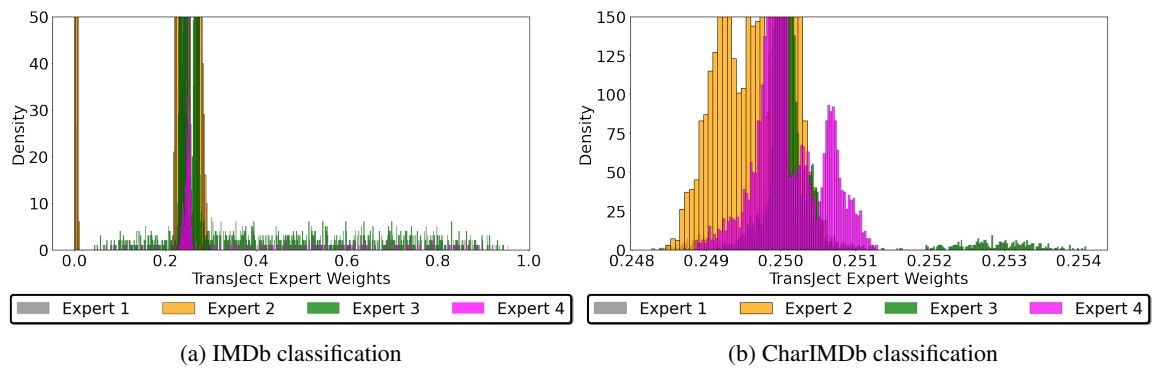

(a) IMDb classification        (b) CharIMDb classification

Figure 7: Expert weights on (a) IMDb and (b) CharIMDb classification tasks. The mode around 0.25 highlights the inherent load-balancing capabilities among the experts.

Scikit-learn[4] that uses $1 - cosine\_similarity$ as the cosine distance. Cosine similarity is the cosine of the angle between two vectors which lies in $[-1, 1]$. Thus the distance lies in $[0, 2]$.

| Model | #Parameters |
|---|---|
| Transformer | 190.1M |
| Transformer with $d_{ff} = 512$ | 95.6M |
| TransJect ($E = 1$) | 66.2M |
| TransJect ($E = 8$) | 286.8M |

Table 5: Comparison of the number of learnable parameters and inference speed on the CharIMDb classification task. The number of parameters is computed for encoders with $L = 6$ and $d = 512$. Transformer uses hidden size $d_{ff} = 2048$ in the feedforward layer, whereas TransJect uses the original encoder hidden dimension to maintain invertibility.

[4]https://scikit-learn.org/stable/modules/generated/sklearn.metrics.pairwise.cosine_distances.html

## C  Analysis

**Connections between entropy and model configuration.** We highlight the connections between activation bounds and total entropy under different model configurations in Figure 6 for TransJect and Transformer, respectively. We observe that more experts in MOE increase activation at the final layer. On the other hand, with a larger dropout for Transformer, the model regularizes more and creates more sparsity, leading to lower activation factors and entropy. Contrary to TransJect, increasing the number of layers of the Transformer increases activation. This behaviour indicates the layer-agnostic property of our model.

**Effectiveness of the expert model.** We observe the individual importance of experts and how they interplay in the mixture model. We illustrate the expert weight distribution in Figure 7, confirming

that the experts are well-balanced and that our model does not require an enforced load balancing. Further, we calculate the entropy of each expert representation to understand how it constituents the final representation at every layer. The expert entropy value of `TransJect` is $-2.2$. Computing an equivalent entropy of different attention heads for the Transformer gives us an entropy of $0.67$. The lower entropy displayed by the mixture of experts highlights the generalization capabilities of the experts. Different experts learn differently but orderly, unlike random and sparse learning by the attention heads of Transformer models.

**Parameter comparison.** We compare the models in terms of total learnable parameters in Table 5. In an equivalent setup, `TransJect` has $50\%$ more learnable parameters than Transformers.