# OpenReview forum: "Manifold-Preserving Transformers are Effective for Short-Long Range Encoding"
_EMNLP/2023/Conference — EMNLP 2023 Findings_

### Official Review · Reviewer_Vm5s · 2023-07-31

**Soundness:** 4

**Excitement:**

3: Ambivalent: It has merits (e.g., it reports state-of-the-art results, the idea is nice), but there are key weaknesses (e.g., it describes incremental work), and it can significantly benefit from another round of revision. However, I won't object to accepting it if my co-reviewers champion it.

**Paper Topic And Main Contributions:**

This paper study how to preserve similarities in transformer and propose a new method for attention. The authors replace the attention mechanism with an orthogonal decomposition $XU\Sigma V$for the input $X$.
 The authors argue that the representation will be injective and low entropy.

**Questions For The Authors:**

- What is the relationship between the preservation of layer-wise distance between a pair of tokens and the classification of sequences into short and long lengths? Does the proposed method gain a better representation? There is still a gap between representation across tokens and sentence representation.
- In line 330, the authors argue that a transformer layer project representations to a lower-dimensional space, does it refer to the multi-head attention operation? Does MoE perform better than single head attention without a lower dimension?  The incorporation of Mixture of Experts (MoE) appears to primarily focus on dimension preservation, whereas there exist numerous alternative techniques that also prioritize the preservation of dimensions.
- How the learnable weight, such as $\alpha$ in equ. 2 and $\lambda$ in equ. 3 show? Could you give more analysis for those weights?
- How to ensure the $W_1$ and $W_2$ matrices to be square orthogonal?
- How the pair-wise distance across token connect to entropy is also not clear.

**Reasons To Accept:**

- The paper is well motivated.
- The paper study an interesting question and conduct extensive experiments.

**Reasons To Reject:**

- The paper is loosely organized  and the correspondence from their theory to their methods is not clear. The paper introduce too many redundant materials, such as line 95 to line 114.
- The paper combine a series of techniques and the truly contribution is not clear. The ablation study for each component should be given to demonstrate their effectiveness.

**Reproducibility:**

4: Could mostly reproduce the results, but there may be some variation because of sample variance or minor variations in their interpretation of the protocol or method.

**Reviewer Confidence:**

3: Pretty sure, but there's a chance I missed something. Although I have a good feel for this area in general, I did not carefully check the paper's details, e.g., the math, experimental design, or novelty.

---

> ### Author Rebuttal · Authors · 2023-08-28
>
> We thank you for the valuable feedback. We address the raised concerns.
>
> **R1.** We request the reviewer to elaborate on the confusion, as it is difficult for us to address this comment without much clarity. Also, lines 95-114 of our paper emphasize the motivation of using entropy to evaluate different encoder models analytically. As stated, entropy is a key metric widely used in cognitive and physical sciences to quantify a state's randomness. We use entropy to evaluate the randomness within different token representations obtained by different encoder models.
>
> **R2.** In this work, we propose a non-normalized orthogonal linear attention and devise an injective variant of the Transformer. Our proposed model TransJect is shown to be Lipschitz continuous with a fixed Lipschitz bound. We observe low activation bound and low entropy for our model, highlighting its effectiveness in encoding short- and long sequences. Our model exhibits significantly lower inference time due to low entropy, highlighting its efficiency in encoding longer sequences.
>
> We have already highlighted the ablation study in Section 4.2 of our paper. Table 1, 2 and 3 highlights the ablation results. We want to clarify that the models -- Orthogonal Transformer and Transformer+ReZero are two ablations we have used to highlight the importance of different components. Orthogonal Transformer uses Orthogonal parameterization but does not use learnable residual. On the other hand, Transformer+ReZero uses learnable residuals but does not use Orthogonal parameterization in any attention or FFN layer. Additionally, we have computed another ablation with only a single expert in the MOE module. This ablation of TransJect highlights the effectiveness of the MOE module. Dropping the MOE module reduces the validation accuracy on the IMDb classification task by $4.7$%. On the CharIMDb classification, MOE contributes an improvement of $1$%.
>
> **Q1.** We hypothesize that restricting the distance between a pair of tokens acts as a regularization, improving the encoder's final predictive performance. We computed the Pearson's correlation between the activation bound of different encoder models (TransJect, Transformer, Orthogonal Transformer and Transformer+ReZero) and the final predictive performance and obtained a correlation score of $-0.81$ with pvalue $0.09$. A similar correlation score of $-0.68$ is obtained on the CharIMDb classification task. These results highlight the negative linear relationship between the final predictive performance and the average activation bound of the encoders. Therefore, we conclude that a tighter activation bound could lead to better predictive performances on short- and long-range encoding tasks.
>
> **Q2.** We would like to clarify that the primary motivation behind introducing MOE is not to preserve dimension but to ensure injectivity within our encoder model. Concatenation operator, usually used in multi-head self-attention, is not an injective operation. Whereas Corollary 1 of our paper shows that the convex sum of different experts leads to an injective mapping.
>
> To highlight the effectiveness of the MOE module, we evaluate an ablation of our model by dropping the MOE module (reducing the number of experts to 1). Dropping the MOE module reduces the validation accuracy on the IMDb classification task by $4.7$% (accuracy $84$%). On the other hand, using only a single head in the original Transformer model reduces its performance on the IMDb classification task by $2.1$% (accuracy $79$%).
>
> **Q3.** We have already provided the analysis on $\alpha$, the residual weight and $\lambda$, the expert weights in Figure 4 and Figure 6 in the Appendix of our paper. Section 5 of the main manuscript also describes the statistical properties of these variables.
>
> **Q4.**  Appendix B.1 of our paper already describes the mechanism we adopted to ensure orthogonality. We use the PyTorch parameterization library to ensure the orthogonality of these learnable parameters. The implementation uses orthogonal maps (e.g. householder mapping) and dynamic trivialization to ensure orthogonality during the forward and backward pass.
>
> **Q5.** We compute Spearman's rank and Pearson correlation to understand the relationship between average activation bound and average entropy. On the IMDb classification task, we observe a Spearman's rank correlation value of $1.0$, with pvalue of $0.001$. The Pearson's correlation also stands at $0.69$. These analyses indicate the positive relationships between the two measures, *i.e.* having a lower activation bound lowers the overall entropy of the model representations.

---

### Official Review · Reviewer_9LuQ · 2023-08-03

**Typos Grammar Style And Presentation Improvements:** NA
**Soundness:** 4

**Excitement:**

4: Strong: This paper deepens the understanding of some phenomenon or lowers the barriers to an existing research direction.

**Missing References:**

NA

**Paper Topic And Main Contributions:**

This paper tries to improve Transformer attention. They have a assumption that the pairwise similarity relationship shall remain similar across layers. Based on this assumption, they propose to replace the dot-product based attention with a orthogonal decomposition (Attn(X) = XU E V).

This ensures injective mappings of token representation across layers.

They show higher performance on several langauge model benchmarks, and give a analysis of multi-head self-attention from statistical physics viewpoint (they have higher empirical activation bounds compared with the proposed TransJect)

**Questions For The Authors:**

As stated above, when do the authors think the assumption (topology shall remain similar across layers) will be true, and when it might fail?

Also, how does the activation bounds relate to eventual performance?

**Reasons To Accept:**

Many analysis in this paper are interesting, and the proposed method to preserve the similarity topology across layer is also interesting and promising. If this method indeed works generally for all the tasks, it will be come a good alternative to current Transformer building blocks.

**Reasons To Reject:**

The key assumption of this paper "the token similarity across layers shall remain the same", might not be correct for all NLP tasks. I think in many situtations, the attention for different layers are trying to learn different semantic relationship and meaning. Some papers also study extracting some syntax tree from attention of different layers: (e.g., https://aclanthology.org/W18-5444/)


I'm interested to hear more from the authors, when their suggested assumption shall work, and at which cases (and tasks) the assumption as well as the proposed method might fail.

**Reproducibility:**

3: Could reproduce the results with some difficulty. The settings of parameters are underspecified or subjectively determined; the training/evaluation data are not widely available.

**Reviewer Confidence:**

3: Pretty sure, but there's a chance I missed something. Although I have a good feel for this area in general, I did not carefully check the paper's details, e.g., the math, experimental design, or novelty.

---

> ### Author Rebuttal · Authors · 2023-08-28
>
> We thank you for the valuable feedback. We address the raised concerns.
>
> **R1.**  We would like to clarify that our intention is not to restrict the distances between a pair of tokens across layers but to bind them within a theoretic bound. Therefore, the model still learns different semantic relationships and topological properties but preserves the overall manifold structure of the embedding space within a specific theoretical bound. This also results in a lower activation bound for our model.
>
> We have computed the Pearson's correlation between the activation bound of different encoder models (TransJect, Transformer, Orthogonal Transformer and Transformer+ReZero) and the final predictive performance and obtained a correlation score of $-0.81$ with $p$-value $0.09$. A similar correlation score of $-0.68$ is obtained on the CharIMDb classification task. These results highlight the negative linear relationship between final predictive performance and the average activation bound of the encoders. Therefore, we conclude that tighter (lower) activation bounds can lead to better predictive performance on short- and long-range encoding tasks.
>
> **Q1.** We think the hypothesis of bounded activation bound should hold in any encoding task. However, the activation bound might change in different semantic and pragmatic tasks, depending upon the necessity of learning different topological properties within texts. For instance, we observe an average activation bound of $1.04$ for TransJect in the IMDb classification task but an activation bound of $1.0$ in CharIMDb classification.
>
> **Q2.** We have computed the Pearson's correlation between the activation bound of different encoder models (TransJect, Transformer, Orthogonal Transformer and Transformer+ReZero) and the final predictive performance and obtained a correlation score of $-0.81$ with $p$-value $0.09$. A similar correlation score of $-0.68$ is obtained on the CharIMDb classification task. These results highlight the negative linear relationship between the final predictive performance and the average activation bound of the encoders. Therefore, we conclude that a tighter activation bound could lead to better predictive performances on short- and long-range encoding tasks.

---

### Official Review · Reviewer_p61L · 2023-08-08

**Soundness:** 3

**Excitement:**

3: Ambivalent: It has merits (e.g., it reports state-of-the-art results, the idea is nice), but there are key weaknesses (e.g., it describes incremental work), and it can significantly benefit from another round of revision. However, I won't object to accepting it if my co-reviewers champion it.

**Paper Topic And Main Contributions:**

This paper proposes a Transformer variant that keeps the distance between elements consistent among layers and heads. This paper achieves its goal by combining orthogonal attention and residual with learnable weight. For experiments, this paper validates the proposed method on a long-range arena, two small-range text classification tasks, and a small language modeling task.

**Questions For The Authors:**

A. Why do we need to keep the value of distance consistent? This paper does not cite related papers or conduct its own studies to prove that.

B. Why is the FFN module called Orthogonal FFN? It seems not to use orthogonal.

C. How much improvement does the MOE bring? Is the MOE specific to your model design?

**Reasons To Accept:**

This paper connects common-used modules and the manifold-preserving abilities.

**Reasons To Reject:**

1. The baselines are weak. This paper does not compare s4 in the LRA benchmark, which achieves much higher results. On text classification and language modeling, this paper also only compares with a few weak baselines.  Moreover, the proposed method uses learnable residual, elu,  and MOE.

2. The contribution brought by this paper is limited.  First, it does not prove that we need to keep the value of distance between elements consistent. Second, orthogonal attention, moe, and residual with learnable weight are both known; what is the unique contribution of this paper?

**Reproducibility:**

3: Could reproduce the results with some difficulty. The settings of parameters are underspecified or subjectively determined; the training/evaluation data are not widely available.

**Reviewer Confidence:**

4: Quite sure. I tried to check the important points carefully. It's unlikely, though conceivable, that I missed something that should affect my ratings.

---

> ### Author Rebuttal · Authors · 2023-08-28
>
> We thank you for the valuable feedback. We address the raised concerns below.
>
> **R1. (weak baselines; s4 not used as a baseline)** We would like to reiterate that the primary motivation of our work is to demonstrate the importance of Lipschitz continuity and injectivity within the Transformer architecture. As the state-space models (S4) do not fall in the Transformer family, we do not compare them with our model. S4 and other state-space models show different inductive biases than TransJect. Therefore, they are not comparable and are excluded from this study.
>
> **R2.** As explained in Section 1 (lines 115-134) of our paper, preserving distance between a pair of tokens within a certain analytical bound ensures that the manifold structure between different tokens is not distorted in larger depths. Our empirical analysis highlights the importance of injectivity and Lipschitz continuity, and the significant reduction in test-time inference time justifies the need for these mathematical properties. Our empirical analysis also highlights the advantages of having low activation bound on the final predictive task. We have computed the Pearson's correlation between the activation bound of different encoder models (TransJect, Transformer, Orthogonal Transformer and Transformer+ReZero) and the final predictive performance and obtained a correlation score of $-0.81$ with $p$-value $0.09$. A similar correlation score of $-0.68$ is obtained on the CharIMDb classification task. These results highlight the negative linear relationship between final predictive performance and the average activation bound of the encoders. Therefore, we conclude that tighter (lower) activation bounds can lead to better predictive performance on short- and long-range encoding tasks.
>
> **Novelty of the paper** We do not claim the novelties with MOE, orthogonal attention or learnable residuals. Our novelty is that we analytically show the Lipschitz continuity of our modified Transformer architecture by unifying these components. Furthermore, reformulating self-attention as $XU\Sigma V$ is novel and essential to ensure the Lipschitz condition. To our knowledge, ours is the first work ensuring injectivity within the Transformer family. Although a few earlier works explored the Lipschitz continuity of Transformers, ours is one of the few studies that have computed the theoretical Lipschitz bound independent of the model configuration.
>
> **Q A.** As explained in Section 1 (lines 115-134) of our paper, preserving distance between a pair of tokens within a certain analytical bound ensures that the manifold structure between different tokens is not distorted in larger depths. Our empirical analysis also highlights the advantages of having low activation bound on the final predictive task. We have computed the Pearson's correlation between the activation bound of different encoder models (TransJect, Transformer, Orthogonal Transformer and Transformer+ReZero) and the final predictive performance and obtained a correlation score of $-0.81$ with $p$-value $0.09$. A similar correlation score of $-0.68$ is obtained on the CharIMDb classification task. These results highlight the negative linear relationship between final predictive performance and the average activation bound of the encoders. Therefore, we conclude that tighter (lower) activation bounds can lead to better predictive performance on short- and long-range encoding tasks.
>
> **Q B.** Orthogonal parameterization ensures a Lipschitz bound of $1$, which is required to ensure that the TransJect Lipschitz bound is layer-independent. Therefore, we use an Orthogonal FFN instead of the vanilla non-parameterized FFN.
>
> **Q C.** MOE is very specific to our proposed model. A convex sum of different experts is required to ensure the injectivity of our encoder. The concatenation operation usually used in the multi-head attention is not injective. Dropping the MOE (i.e., the number of experts is 1) drops the validation accuracy on the IMDb classification task by $4.7%$%. On the CharIMDb classification, MOE contributes an improvement of $1$%.

---

### Meta-Review · Area_Chair_KHbr · 2023-09-19

**Recommendation:** 3

**Metareview:**

This paper introduces a Transformer variant that maintains the distance between elements consistent across layers and heads. The paper achieves this by combining orthogonal attention, residual, and learnable weight. The method is validated on LRA and few small datasets.

Pros:
- The paper proposes to combine orthogonal attention with residual networks and analytically show the Lipschitz continuity of their modified Transformer architecture. Which is an interesting idea.

Cons:
- It is unclear why “preserving distance between a pair of tokens within a certain analytical bound” is actually useful or needed. This makes the objective of the paper unclear.
- In the absence of clearly motivated objectives, it is unclear as to why Lipschitz continuity is required (as shown by the authors for their model).
- Actual contribution of the paper is unclear
- The reason behind a >5x improvement of speed while 50% more parameter is not clear and not analyzed in the paper.

---

### Decision · Program_Chairs · 2023-10-07

**Decision:**

Accept-Findings

**Comment:**

This paper introduces a Transformer variant that maintains the distance between elements consistent across layers and heads. The paper achieves this by combining orthogonal attention, residual, and learnable weight. The method is validated on LRA and few small datasets.

Pros:
- The paper proposes to combine orthogonal attention with residual networks and analytically show the Lipschitz continuity of their modified Transformer architecture. Which is an interesting idea.

Cons:
- It is unclear why “preserving distance between a pair of tokens within a certain analytical bound” is actually useful or needed. This makes the objective of the paper unclear.
- In the absence of clearly motivated objectives, it is unclear as to why Lipschitz continuity is required (as shown by the authors for their model).
- Actual contribution of the paper is unclear
- The reason behind a >5x improvement of speed while 50% more parameter is not clear and not analyzed in the paper.